# Additional Anomalies in Children with Gastroschisis and Omphalocele: A Retrospective Cohort Study

**DOI:** 10.3390/children10040688

**Published:** 2023-04-05

**Authors:** Adinda G. H. Pijpers, Cunera M. C. de Beaufort, Sanne C. Maat, Chantal J. M. Broers, Bart Straver, Ernest van Heurn, Ramon R. Gorter, Joep P. M. Derikx

**Affiliations:** 1Department of Pediatric Surgery, Emma Children’s Hospital Amsterdam UMC, Location University of Amsterdam, Meibergdreef 9, 1105 AZ Amsterdam, The Netherlands; 2Amsterdam Gastroenterology and Metabolism Research Institute, 1105 AZ Amsterdam, The Netherlands; 3Amsterdam Reproduction and Development Research Institute, 1105 AZ Amsterdam, The Netherlands; 4Department of Pediatrics, Emma Children’s Hospital Amsterdam UMC, Location University of Amsterdam, Meibergdreef 9, 1105 AZ Amsterdam, The Netherlands; 5Department of Pediatric Cardiology, Emma Children’s Hospital Amsterdam UMC, Location University of Amsterdam, Meibergdreef 9, 1105 AZ Amsterdam, The Netherlands

**Keywords:** abdominal wall defect, gastroschisis, omphalocele, additional anomalies, cardiac anomalies

## Abstract

Background: Congenital abdominal wall defects might be associated with other anomalies, such as atresia in gastroschisis and cardiac anomalies in omphalocele patients. However, in the current literature, an overview of these additional anomalies and potential patient-specific risk factors is missing. Therefore, we aimed to assess the prevalence of associated anomalies and their patient-specific risk factors in patients with gastroschisis and omphalocele. Methods: A mono-center retrospective cohort study between 1997 and 2023 was performed. Outcomes were the presence of any additional anomalies. Risk factors were analyzed via logistic regression analysis. Results: In total, 122 patients were included, of whom 82 (67.2%) had gastroschisis, and 40 (32.8%) had omphalocele. Additional anomalies were identified in 26 gastroschisis patients (31.7%) and in 27 omphalocele patients (67.5%). In patients with gastroschisis, intestinal anomalies were most identified (n = 13, 15.9%), whereas, in patients with omphalocele, cardiac anomalies were most identified (n = 15, 37.5%). Logistic regression showed that cardiac anomalies were associated with complex gastroschisis (OR: 8.5; CI-95%: 1.4–49.5). Conclusions: In patients with gastroschisis and omphalocele, intestinal and cardiac anomalies were most identified, respectively. Cardiac anomalies were found to be a risk factor for patients with complex gastroschisis. Therefore, regardless of the type of gastroschisis and/or omphalocele, postnatal cardiac screening remains important.

## 1. Introduction

Congenital abdominal wall defects comprise a large spectrum of malformations and affect approximately 1 per 2000 live births yearly [1]. Two of the most common types are gastroschisis and omphalocele [2,3]. Gastroschisis and omphalocele are often identified through prenatal screening, and both mostly require surgical correction in childhood. However, the prevalence, embryogenesis, clinical characteristics, and the relation with potential associated anomalies of both congenital abdominal wall defects vary widely [4].

Both gastroschisis and omphalocele can occur in isolation but might also be associated with additional anomalies [1,4]. Additional intestinal anomalies are well-known in patients with gastroschisis, as atresia at different levels of the intestinal tract can complicate the condition, defining gastroschisis in simple (i.e., without atresia) and complex gastroschisis (i.e., with atresia). In patients with omphalocele, cardiac anomalies are a commonly related anomaly, and, therefore, routine screening has already been recommended in current clinical practices [5,6,7]. Unfortunately, there is a lack of knowledge of other potential additional anomalies that might occur in this patient group [7,8]. In addition, no patient-specific risk factors for complex gastroschisis or cardiac anomalies in patients with omphalocele have yet been identified or described.

It is of great importance to identify patients that are at risk for having additional anomalies, as these additional anomalies might also have treatment consequences on their own or might influence the treatment for the congenital wall defect. For instance, cardiac or intestinal anomalies potentially require acute surgical correction and are of importance for the anesthesiologist. In addition, if patients are suspected of having complex genetic disorders, consultation with a clinical geneticist is of importance. Therefore, the aim of this study is threefold. First, to assess the prevalence of additional anomalies in children with congenital abdominal wall defects (i.e., gastroschisis and omphalocele). Second, to identify patient-specific risk factors for these additional anomalies. Third, to evaluate differences in the prevalence of additional anomalies between gastroschisis and omphalocele patients.

## 2. Materials and Methods

### 2.1. Study Design and Patient Population

This was a retrospective cohort study designed in accordance with the Strengthening the Reporting of Observational Studies in Epidemiology (STROBE) guidelines [9]. Data were retrieved from a retrospectively maintained database using IBM SPSS Statistics for Windows, Version 28.0 (IBM Corp., Armonk, NY, USA). This database included newborns with gastroschisis or omphalocele who were born between June 1997 and January 2023 in Amsterdam University Medical Centers (Amsterdam UMC), former Academic Medical Center (AMC), and VU University Medical Center (VUMC), a tertiary referral center in the Netherlands. Eligible for inclusion were all patients from the retrospectively maintained database. Follow-up was calculated from date of birth to latest hospital visit or censored at death.

### 2.2. Ethics

This study was reviewed by the medical ethical commission and was not subject to the WMO statement (ref. no. W18_233 #18.278, July 2018). Regarding the primary Emma Children’s hospital database, written information was provided to parents or legal guardians for all identified patients with congenital abdominal wall defects (i.e., gastroschisis and omphalocele), including a letter of objection. In case of objection, patients were removed from the database (n = 3).

### 2.3. Data Extraction

Data on patient characteristics, type of congenital abdominal wall defect (i.e., gastroschisis and omphalocele), and additional anomalies (i.e., cardiovascular, vertebral, renal, urogenital tract, eye, ear and neck, esophageal, respiratory, anal, limb, cleft lip and palate, nervous system, and genetic syndromes) were extracted. Furthermore, data on additional testing (e.g., spinal ultrasound and X-ray, cardiac and renal ultrasound, additional testing performed on the indication, genetic counseling, and genetic testing) were extracted from medical records by two independent researchers (A. P., Cd. B., both MDs) according to predefined data collection forms. In case of discrepancies, another author (R. G. or J. D., both pediatric surgeons) was consulted for the final verdict. Data validation was done by checking approximately 20% of the entered records by another author (S. M., also an MD). In case of discrepancies in more than 1% of a record, the complete record was checked.

### 2.4. Postnatal Routine Screening Protocol

Since 2018, a clinical care pathway has been implemented in the Amsterdam UMC, including routine screening for additional cardiac anomalies with cardiac ultrasound in children with omphalocele. For patients with gastroschisis, no standard screening protocol was in place, and additional testing was done on indication and at the treating physicians’ discretion. In case anomalies were suspected during clinical assessment, additional imaging studies were undertaken to confirm or rule out the suspected diagnosis.

### 2.5. Definitions

Congenital abdominal wall defects were defined as gastroschisis and omphalocele. For this study, gastroschisis was further classified into simple (i.e., without intestinal atresia) and complex gastroschisis (i.e., with intestinal atresia (jejunal, ileal, and colonic atresia)). Omphalocele was further classified into minor (i.e., abdominal wall defect ≤ 5 cm and/or without liver in omphalocelic sac) and giant omphalocele (i.e., abdominal wall defect ≥ 5 cm and/or with liver in omphalocelic sac).

Additional anomalies were divided into different organ systems (i.e., cardiovascular, vertebral, renal, urogenital tract, eye, ear, neck, esophageal, gastro-intestinal, limbs, cleft, lip and palate, nervous system, and genetic disorders) and were classified as reported in the patients’ medical charts accordingly. In case of uncertainties regarding the classification of additional anomalies and symptoms, an expert panel consisting of a pediatrician, pediatric cardiologist, radiologist, urologist, neurologist, clinical geneticist, and pediatric surgeon was consulted. If no information was available in the medical record on whether or not screening for additional anomalies was performed, it was classified as missing.

### 2.6. Outcomes

The primary outcome was the number of patients with congenital abdominal wall defects (i.e., gastroschisis and omphalocele) with additional anomalies according to the type of abdominal wall defect.

Secondary outcomes were to determine differences in the prevalence of additional anomalies between gastroschisis and omphalocele patients, as well as to determine patient-specific risk factors for these additional anomalies.

### 2.7. Statistical Analyses

Statistical analysis was conducted using SPSS Version 28.0. Descriptive statistics were used for the analysis of baseline characteristics. These were reported as proportions and percentages for binary or categorical variables and as mean with standard deviation (SD) or as median with interquartile range (IQR) for continuous variables as appropriate. To test differences in categorical data, the Chi-square test and, additionally, Fisher’s Exact test were used in case no assumptions were met.

Uni- and multivariate logistic regression were used to identify potential associations between cardiac anomalies in complex gastroschisis as well as in omphalocele patients. The selection of input variables for univariable regression analysis was based on additional anomalies identified in the previous literature and after consensus within the research team. Backward Wald selection was used to select variables using the standard *p* = 0.10 for variable removal. Outcomes were reported as odds ratio (OR) with a corresponding 95% confidence interval (95% CI). Confounding (increase in B-coefficient of >10%) and effect modification (significant interaction term) assessments were performed. Additionally, the adjusted R-squared was reported to show the proportion of the variance in the occurrence of complex gastroschisis and cardiac anomalies in patients with omphalocele explained by the model. A *p*-value of <0.05 was considered statistically significant. Missing or unknown data were described.

## 3. Results

### 3.1. Participants

In total, 122 patients were identified, of whom all patients were eligible for inclusion in this study. The cohort comprised 82 patients (67.2%) with gastroschisis and 40 patients (32.8%) with omphalocele, with a median age at the latest follow-up of 23.0 months (IQR 4.0–55.3). In 11 of the gastroschisis patients (13.4%), an atresia was found, resulting in complex gastroschisis. Giant omphalocele was present in 10 neonates with omphalocele (25.0%). Closure of the gastroschisis was a primary closure in 46 neonates (56.1%), silo placement in 29 (35.4%), and 7 patients were treated with temporary enterostomy (8.5%). Most omphalocele patients underwent primary closure (n = 33, 82.5%), but also epithelialization (n = 6, 15.0%) and enterostomy (n = 1, 2.5%) were performed as omphalocele treatments. Eight patients (6.6%) died during the study period due to ongoing sepsis (n = 6, at ages 29, 58, 61 days, 4.3 and 8.3 months, and 14.7 years) and severe neurological outcome (n = 1, aged 27 days). The cause of death was not available in the medical chart of one patient. An overview of baseline characteristics is shown in Table 1.

### 3.2. Postnatal Screening

Genetic screening was performed at the clinical geneticist’s discretion (i.e., karyotyping, array CGH analysis, Sanger sequencing, or next-generation sequencing) in 58 gastroschisis (70.7%) and 32 omphalocele (80.0%) patients. Sixty-two gastroschisis (75.9%) and 36 omphalocele (90.0%) patients underwent neurological screening with cerebral ultrasound, and in 58 gastroschisis (70.7%) and 32 omphalocele (80.0%) patients, vertebral screening was performed with vertebral x-ray and spinal cord ultrasound. In addition, in 45 gastroschisis (54.9%) and 31 omphalocele (77.5%) patients, cardiac screening was performed with cardiac ultrasound, and in 61 gastroschisis (74.4%) and 34 (85.0%) omphalocele patients, renal screening was performed with renal ultrasound. In total, 63 gastroschisis (76.8%) and 36 omphalocele (90.0%) patients underwent urogenital screening with physical examination, abdominal ultrasound, voiding cysto-urography, or urodynamic research.

### 3.3. Additional Anomalies

Additional anomalies were identified in 53 patients (43.3%), of whom 25 (47.2%) had a single additional anomaly, and 28 (52.8%) had multiple anomalies (i.e., ≥2 anomalies in different organ systems). In 26 patients with gastroschisis (31.7%), additional anomalies were identified, of whom 17 had a single (65.4%), and 9 had multiple anomalies (34.6%). In 27 omphalocele patients (67.5%), additional anomalies were found, of whom 8 had a single and 19 had multiple anomalies (70.1%). An overview of all identified additional anomalies can be found in Table 2.

In patients with gastroschisis, intestinal anomalies were identified in 13 patients (13.4%) with 15 different types of intestinal anomalies. In 11 patients (13.4%) with complex gastroschisis, atresia was located in the jejunum (n = 2), ileum (n = 4), colon (n = 3), and multiple locations (n = 2). Other intestinal anomalies identified were Meckel’s diverticulum (n = 2), meconium ileus (n = 1), and choledochal cyst (n = 1). There was no significant difference in the prevalence of intestinal anomalies compared to patients with omphalocele (*p* = 0.260).

In patients with omphalocele, cardiac anomalies were most frequently identified (n = 15, 38.5%). The majority of these cardiac anomalies existed of septal defects, including atrial septal defects (n = 5, 31.3%) and ventricular septal defects (n = 5, 31.3%). Nine patients (30.0%) with a minor omphalocele had a cardiac anomaly, whereas, in six patients (60.0%) with a giant omphalocele, cardiac anomalies were identified. Anomalies that were significantly more frequently identified in patients with omphalocele compared to gastroschisis were urogenital (n = 10, *p* = 0.032), cleft, lip and palate (n = 7, *p* = 0.002), genetic anomalies (n = 7, *p* < 0.001), and anomalies of the limbs (n = 4, *p* = 0.039).

### 3.4. Patient-Specific Risk Factors

To facilitate the differentiation between simple and complex gastroschisis directly postpartum, patient-specific risk factors were assessed for patients with complex gastroschisis. Univariate logistic regression showed the presence of cardiac anomalies (OR: 8.5; CI-95% 1.5–49.4) as a significant risk factor for patients with complex gastroschisis (Table 3A). This model has a Nagelkerke R2 of 11%.

Since cardiac anomalies were the most frequently identified additional anomalies in patients with omphalocele, patient-specific risk factors were determined for this group. Multivariate logistic regression showed male sexe (OR: 1.4; CI-95% 0.4–5.5) and giant omphalocele (OR: 3.5; CI-95% 0.8–15.5) not being significant risk factors and could, therefore, not be included in the model (Table 3B).

## 4. Discussion

Yearly, the prevalence of gastroschisis and omphalocele in the general population is 2.76–3.63 per 10.000 births, respectively. With 170,000–18,000 live births in the Netherlands in the last years, the incidence of gastroschisis and omphalocele is approximately 50 new patients per year, of which we see five per year in our center on average [10]. In our cohort of 122 patients (gastroschisis (n = 82) and omphalocele (n = 40)), additional anomalies were identified in almost half of the patients (43%). Thirteen patients with gastroschisis had 15 associated anomalies, of which intestinal atresias were mostly identified (n = 11), resulting in a classification of complex gastroschisis, followed by Meckel’s diverticulum (n = 2) and meconium ileus (n = 1). With univariate logistic regression, we were able to identify cardiac anomalies as a patient-specific risk factor for complex gastroschisis (OR: 8.5; CI-95% 1.5–49.4). Cardiac anomalies were most frequently identified in patients with omphalocele (minor (n = 9/30) and giant (n = 6/10)) and were significantly more often present in patients with omphalocele compared to gastroschisis (*p* ≤ 0.001), as expected. In addition, urogenital (n = 10, *p* = 0.032), cleft, lip and palate (n = 7, *p* = 0.002), genetic anomalies (n = 7, *p* < 0.001), and anomalies of the limbs (n = 4, *p* = 0.039) were significantly more present in patients with omphalocele. No patient-specific risk factors could be identified for cardiac anomalies in patients with omphalocele.

Similar to previous studies, we were able to show gastroschisis being most often associated with intestinal anomalies [11]. The primary concern for gastroschisis patients is whether the abdominal wall defect is simple or complex. In our cohort of gastroschisis patients, 15% had complex gastroschisis, which is comparable to the prevalence reported in the literature [2]. Furthermore, mortality rates in patients with complex gastroschisis are significantly higher compared to patients with simple gastroschisis (complex 16.7% versus simple 2.2%) [11]. Therefore, it is of great importance to recognize complex gastroschisis as early as possible so treatment can quickly be initiated (e.g., surgery for atresia, mal-rotation, and perforation). Although gastroschisis can be diagnosed on prenatal ultrasound from 12 weeks gestation, antenatal and postnatal differentiation between simple and complex gastroschisis remains challenging [11]. Several studies aimed to determine antenatal risk factors that could indicate whether atresia was present, but these risk factors remain inconclusive [12]. Patient-specific risk factors were never examined and may aid in predicting the presence of complex gastroschisis. This study showed that the presence of cardiac anomalies is a significant risk factor for complicated gastroschisis, which suggests that the probability of intestinal atresia should be given additional consideration in case of the presence of a cardiac anomaly. Kunz et al. suggested that detailed antenatal and/or postnatal cardiac evaluations are indicated in fetuses identified with gastroschisis due to the significantly higher incidence of congenital heart defects (*p* = 0.014) in patients with complex gastroschisis [13]. These cardiac evaluations may result in the identification of patients at higher risk for complex gastroschisis. An explanation for this higher incidence of cardiac anomalies is possibly due to placental maldevelopment. Delayed villous maturation, which is linked to gastroschisis and affects placental blood flow, may be the root of this maldevelopment [14]. It is known that placental blood flow might contribute to early cardiac hemodynamics [15,16]. This suggests that these placental changes possibly contribute to cardiac anomalies in these patients.

Cardiac anomalies were identified in 39% of the omphalocele patients. Postnatal cardiac evaluation in omphalocele patients is standard care in the Netherlands due to the high risk of associated cardiac anomalies, as cardiac anomalies are present in 15–50% of children with omphalocele [17]. It is important to assess potential cardiac anomalies preoperatively since the early reduction of the omphalocele can cause an alteration in preload and afterload in patients with cardiac anomalies, which may result in hemodynamic instability [18]. This study, however, was not able to identify patient-specific risk factors for cardiac anomalies in omphalocele patients. Hence, postnatal cardiac screening remains of importance for all omphalocele patients, regardless of type. Not only is a cardiac screening of importance for omphalocele patients but also genetic screening plays an important role in the workup. Mostly small omphalocele are associated with an abnormal karyotype, the most common trisomy 13/18/21, Beckwith–Wiedemann syndrome, and pentalogy of Cantrell [18]. Our study reported an incidence of genetic anomalies of 17.5% (n = 7), of which five patients were diagnosed with Beckwith–Wiedemann syndrome. The association with this syndrome possibly explains the higher prevalence of cleft, lip, and palate anomalies in patients with omphalocele.

It is important to consider this study’s strengths and limitations when interpreting the findings. For patients with gastroschisis and omphalocele, this study is the first to identify risk factors for complex gastroschisis and cardiac anomalies in patients with omphalocele. However, there is a number of limitations to this study, as there are to all retrospective cohort studies, most importantly, information and registration bias, particularly given the long timeframe in which the study was executed. In addition, not all patients underwent full screening, or screening was not properly documented in the available medical records, which may have led to underreporting of the actual number of additional anomalies present in this cohort. In addition, this study does not comprise prenatal data such as prenatal incidence and detection rate and termination percentage. Additionally, information regarding prenatal cardiac and genetic screening is lacking in this study, even though this plays an important role in the diagnosis of additional anomalies. Further studies should take all these outcomes into account. Furthermore, discussion remains on accurate definitions of minor and giant omphalocele, and in our center, we may have used different interpretations of these terms compared to other studies. In contrast to the body of existing literature, this may result in under- and/or over-reporting of the factual number of minor, respectively, giant omphalocele patients.

## 5. Conclusions

This study provides an overview of all identified additional anomalies in patients with gastroschisis or omphalocele. In almost half of the patients, additional anomalies were identified, of which 47.2% were single anomalies, and 52.8% were multiple anomalies. Intestinal anomalies, such as atresia, Meckel’s diverticulum, and meconium ileus, were found most common in patients with gastroschisis, and cardiac anomalies, including septal defects, such as atrial or ventricular septal defect, were most often found patients with omphalocele. This was followed by urogenital anomalies, such as cryptorchidism and hypospadias, and genetic anomalies, which mainly included Beckwith–Wiedemann syndrome. The presence of cardiac anomalies was found to be a risk factor for complex gastroschisis compared to simple. On the other hand, we did not identify patient-specific risk factors for omphalocele patients with cardiac anomalies. Finally, a comparison between the two groups of abdominal wall defects showed that patients with an omphalocele had more cardiac, urogenital, cleft, lip and palate, and limb anomalies compared to gastroschisis patients. Therefore, postnatal cardiac screening remains of importance for all patients with congenital abdominal wall defects, regardless of type. Prenatal cardiac and genetic screening may be beneficial in the diagnostic process of both gastroschisis and omphalocele patients and should be further investigated.

## Figures and Tables

**Table 1 children-10-00688-t001:** Baseline characteristics.

		Total, n = 122	Gastroschisis, n = 82	Omphalocele, n = 40
		n (%)	n (%)	n (%)
Sexe				
Male		68 (55.7)	47 (57.3)	21 (52.5)
Female		54 (44.3)	35 (42.7)	19 (47.5)
Premature birth		58 (47.5)	46 (56.1)	12 (30.0)
Multiple birth		5 (4.1)	1 (1.2)	4 (10.0)
Birth route				
C-section		37 (30.3)	25 (30.5)	12 (30.0)
Vaginal		85 (69.7)	57 (69.5)	28 (70.0)
Antenatal diagnosis		103 (84.4)	76 (92.7)	27 (67.5)
Additional anomaly		53 (43.3)	26 (31.7)	27 (67.5)
Single		25 (47.2)	17 (65.4)	8 (30.6)
Multiple		28 (52.8)	9 (34.6)	19 (70.4)
Died in follow-up		8 (6.6)	6 (7.3)	2 (5.0)
Type	Simple	NA	71 (86.6)	NA
	Complex	NA	11 (13.4)	NA
	Minor	NA	NA	30 (75.0)
	Giant	NA	NA	10 (25.0)
		median (IQR)	median (IQR)	median (IQR)
Birthweight, gram		2635.0 (2337.5–3100.0)	2462.5 (2216.3–2913.8)	3077.5 (2596.3–3527.5)
Age at surgery, days		0.0 (0.0–1.0)	0.0 (0.0–1.0)	1.0 (0.0–1.0)
In hospital stay, days		27.0 (11.0–48.0)	34.0 (20.8–53.0)	11.0 (5.0–25.3)
Age at follow-up, months		23.0 (4.0–55.3)	18.0 (4–55.5)	33.0 (7.3–55.8)

**Table 2 children-10-00688-t002:** Additional anomalies in patients with gastroschisis and omphalocele.

	Gastroschisis	Omphalocele	*p*-Value
	n (%) *	(%) *	
**Genetic anomalies**	**0 (0.0)**	**7 (17.5)**	**<0.001**
Beckwith–Wiedemann syndrome	0	5	
Down syndrome	0	1	
Pierre–Robin sequence	0	1	
Missing	24 (29.3)	8 (20.0)	
**Neurological anomalies**	**5 (6.1)**	**3 (7.5)**	**0.716**
Hydrocephalus	2	0	
Encephalopathy	1	0	
Microcephaly	0	1	
Germinolytic cysts	1	0	
Hypotonic extremities	0	1	
Periventricular leucomalacia	0	1	
Lesion in thalamus	1	0	
Missing	24 (29.3)	8 (20.0)	
**Vertebral anomalies**	**2 (2.4)**	**3 (7.5)**	**0.329**
Hip dysplasia	1	1	
Spina bifida	0	1	
Sacral dimple	0	1	
Extra lumbal vertebra	1	0	
Missing	37 (45.1)	9 (22.5)	
**Cardiac anomalies**	**6 (7.3)**	**15 (38.5)**	**<0.001**
Atrial septal defect	0	5	
Ventricular septal defect	0	5	
Persistent ductus arteriosis	1	2	
Pulmonary artery stenosis	2	1	
Aortic valve stenosis	0	1	
Supraventricular tachycardia	1	0	
Dilatation of the aorta ascendens	1	0	
Atrial flutter	1	0	
Dextrocardia	0	1	
Missing	37 (45.1)	9 (22.5)	
**Renal anomalies**	**6 (7.3)**	**6 (15.0)**	**0.205**
Cyst in the kidney	1	2	
Dysplastic kidney	1	1	
Dilatation of pyelum	2	1	
UPJ-stenosis	1	0	
Hydronephrosis	1	0	
Dilatation of the ureter	0	2	
Missing	21 (25.6)	6 (15.0)	
**Urogenital anomalies**	**8 (9.8)**	**10 (25.0)**	**0.032**
Cryptorchidism	3	3	
Hypospadia	1	2	
Urethral stricture	1	1	
Inguinal hernia	1	1	
Ovarial cyst	1	0	
Hydrocele	1	1	
Neurogenic bladder	0	1	
Vesico ureteral reflux	0	1	
Missing	19 (23.2)	45 (56.1)	
**Eye, ear and neck anomalies**	**1 (1.2)**	**1 (2.5)**	**0.55**
Conductive hear loss	1	1	
**Cleft, lip and palate anomalies**	**1 (1.2)**	**7 (17.5)**	**0.002**
Macroglossia	0	2	
Choanal atresia	0	1	
Micrognatia	0	2	
Palatoschisis	0	1	
Subglottic cyst	0	1	
Trismus	1	0	
**Esophageal anomalies**	**1 (1.2)**	**1 (2.5)**	**0.55**
Esophageal atresia	0	1	
Para-esophageal hernia	1	0	
**Respiratory anomalies**	**0 (0.0)**	**2 (5.0)**	**0.106**
Bronchomalacia	0	1	
Pulmonary hypoplasia	0	1	
**Intestinal anomalies**	**13 (15.9)**	**3 (7.5)**	**0.26**
Atresia	11	0	
Meckels diverticulum	2	3	
Meconium ileus	1	0	
Choledochus cyst	1	0	
**Limb anomalies**	**1 (1.2)**	**4 (10.0)**	**0.039**
Anomaly of the thumb	0	1	
Anomaly of the feet	0	2	
Proximal femur deficiency	0	1	
Athrogryposis multiplex congenita	1	0	
**Anal anomalies**	**0 (0.0)**	**0 (0.0)**	**-**
**Total anomalies**	**26/82 (31.7)**	**27/40 (67.5)**	

* Percentages were calculated from screened patients.

**Table 3 children-10-00688-t003:** (A) Univariate logistic regression for the association between baseline characteristics and gastroschisis. (B) Multivariate logistic regression for the association between baseline characteristics and omphalocele.

(**A**)		
	**OR (95% CI)**	***p*-Value**
Cardiac anomalies		
Not present	Ref	
Present	8.500 (1.462–49.406)	0.017
(**B**)		
	**OR (95% CI)**	***p*-value**
Sex		
Female	Ref	
Male	1.446 (0.381–5.492)	0.588
Type		
Simple omphalocele	Ref	
Giant omphalocele	3.493 (0.785–15.541)	0.101

## Data Availability

Data is unavailable due to privacy restrictions.

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
