# Peer review of "Additional Anomalies in Children with Gastroschisis and Omphalocele: A Retrospective Cohort Study"

_children, 2023, doi:10.3390/children10040688_

Round 1

Reviewer 1 Report

Dear authors,

this is a well designed and performed cohort study on gastroschisis and omphalocele in a group of 122 patients over 26 years between 1997-2023 according to STROBE recommendations for observational studies.

You concentrated on additional defects present with the two abdominal wall defects and found clinically significant correlations with cardiac defects.

In order to improve your manuscript you may want to add following information:

1/ Incidence of gastroschisis (G) and omphalocele (O) in a/ general human population, b/ in your country, c/ in your hospital/cohort.

2/ 122 pts/26 yrs gives only ca 5 cases annually, what was the prenatal incidence and detection rate and termination percantage when G or O was diagnosed in your hospital.

3/ You described the postnatal genetic screening, what was the rate of prenatal genetic testing? Was it amniocentesis or cell-free DNA or both?

4/ In line 214 and 215 you duplicated writing Discussion.

5/ In conclusions you may advocate also the importance of PRENATAL cardiac and genetic invasive screening as a gold standard apart from postnatal one.

Kind regards.

Author Response

Response to the comments of the Reviewers

We thank the reviewers for the critical appraisal of our work and the positive remarks on our study.

Reviewer #1: This is a well-designed and performed cohort study on gastroschisis and omphalocele in a group of 122 patients over 26 years between 1997-2023 to STROBE recommendations for observational studies. In order to improve your manuscript you may want to add following information:

  1. Incidence of gastroschisis (G) and omphalocele (O) in a/ general human population, b/ in your country, c/ in your hospital cohort

Reaction: We thank the reviewer for this suggestion and added our discussing section accordingly (P. 8 lines 216-219).

  1. 122pts/26yrs gives only a 5 cases annually, what was the prenatal incidence and detection rate and termination percentage when G or O was diagnosed in your hospital?

Reaction: Unfortunately, prenatal data was beyond the scope of our database and we do not have this information. We agree with the reviewer that prenatal incidence, detection rate and termination percentage are of importance but we cannot provide this data. We have incorporated this in the limitation section of the discussion (P.9-10 286-298).

  1. You described postnatal genetic screening, what was the rate of prenatal genetic testing? Was it amniocentesis or cell-free DNA or both?

Reaction: We do not have these prenatal data of our babies with congenital abdominal wall defects since these results are noted in the patient file of the mother in our country and due to privacy legislation we were not able to search their patient file. We added a section to the limitations of our discussion addressing this important clinical item (P.9-10 286-298).

  1. In line 214 and 215 you duplicated writing Discussion.

Reaction: We thank the reviewer for pointing out the duplicated word and deleted it (P. 8 lines 215).

  1. In conclusions you may advocate also the importance of PRENATAL cardiac and genetic invasive screening as a gold standard apart from postnatal one.

Reaction: We thank the reviewer for this suggestion. Since we did not analyse prenatal cardiac and genetic invasive screening, we cannot advice on prenatal cardiac and genetic screening as a gold standard, but we do advocate inclusions of these parameters in future studies. Therefore we addressed this point in our discussion on P. 10 lines 308-310.

Reviewer 2 Report

The manuscript is well written. I agree that intestinal and cardiac anomalies are more often identified in patients with gastroschisis and omphalocele. This data is similar to my database. Cardiac abnormalities were found to be a risk factor for patients with complex gastroschisis, and postnatal cardiac screening is strongly advised.  The authors should mention and list the type of cardiac abnormalities according to SSA and discuss the role of placental changes as it has been reported in the literature.

Author Response

Response to the comments of the Reviewers

We thank the reviewers for the critical appraisal of our work and the positive remarks on our study.

Reviewer #2: The manuscript is well written. I agree that intestinal and cardiac anomalies are more often identified in patients with gastroschisis and omphalocele. This data is similar to my database. Cardiac abnormalities were found to be a risk factor for patients with complex gastroschisis, and postnatal cardiac screening is strongly advised.

  1. The authors should mention and list the type of cardiac abnormalities according to SSA and discuss the role of placental changes as it has been reported in the literature.

Reaction: We thank the reviewer for the valuable suggestion regarding the role of placental changes in our manuscript. As suggested by the reviewer, we added a section to the discussion using recently published studies about the role of placental changes in patients with gastroschisis for cardiac anomalies (P. 9 lines 254-259).

All patients in our cohort with a cardiac anomaly in our cohort fell under group B of the SSA classification since surgery was not indicated and they were under the age of 12 years. To avoid confusion by the readers we didn’t apply another classification in our manuscript.
